

# Validation of a lookup-table approach to modeling turbine fatigue loads in wind farms under active wake control

Hector Mendez Reyes[1], Stoyan Kanev[1], Bart Doekemeijer[2], and Jan-Willem van Wingerden[2]

[1]TNO, ECN-TNO, Westerduinweg 3, 1755 LE, Petten, the Netherlands
[2]TU-Delft, DCSC, Mekelweg 2, 2628 CD Delft, the Netherlands

**Correspondence:** S. Kanev (stoyan.kanev@tno.nl)

**Abstract.** Wake redirection is an active wake control (AWC) concept that is known to have a high potential for increasing the overall power production of wind farms. Being based on operating the turbines with intentional yaw misalignment to steer wakes away from downstream turbines, this control strategy requires careful attention to the loads implications. However, the computational effort required to perform an exhaustive analysis of the site-specific loads on each turbine in a wind farm is unacceptably high due to the huge number of aeroelastic simulations required to cover all possible inflow and yaw conditions. To reduce this complexity, a practical loads modeling approach is based on "gridding", i.e., performing simulations only for a subset of the range of environmental and operational conditions that can occur. Based on these simulations, a multi-dimensional lookup table (LUT) can be constructed containing the fatigue and extreme loads on all components of interest. Using interpolation, the loads on each turbine in the farm can the be predicted for the whole range of expected conditions. Recent studies using this approach indicate that wake redirection can increase the overall power production of the wind farm and at the same time decrease the lifetime fatigue loads on the main components of the individual turbines. As the present level of risk perception related to operation with large yaw misalignment is still substantial, it is essential to increase the confidence level in this LUT-based loads modeling approach to further derisk the wake redirection strategy. To this end, this paper presents the results of a series of studies focused on the validation of different aspects of the LUT loads modeling approach. These studies are based on detailed aeroelastic simulations, two wind tunnel tests, and a full-scale field test. The results indicate that the LUT approach is a computationally efficient methodology for assessing the farm loads under AWC, which achieves generally good prediction of the load trends.

## 1 Introduction

When wind turbines are grouped into wind farms, they affect each other's performance through their wakes. In the wake, wind turbines experience a decreased wind velocity and increased turbulence. For this reason, waked turbines will produce less power at below rated wind speeds and suffer increased fatigue loading. Below rated, the conventional "greedy" control approach aims at maximizing the power capture for each turbine, thereby disregarding the interactions between the turbines through their wakes. This approach is not optimal with respect to the total power production of the whole wind farm. AWC is an





approach to operate the turbines cooperatively with the goal of mitigating the wake effects to maximize the power production of the whole farm, while at the same time trying to reduce the fatigue loading on the turbines (Kanev et al., 2018).

There are two concepts to AWC. The first concept, known as induction control, adjusts the axial induction of the windward turbines below their optimum for power production in order to reduce the velocity deficit and turbulence in the wake (Corten and

5 Schaak, 2004; Boorsma, 2015; Annoni et al., 2016; Campagnolo et al., 2016). The second strategy, known as wake redirection (or yaw-based AWC), consists of redirecting the wakes aside from the downstream turbines by operating the upwind turbines at a yaw misalignment (Corten et al., 2004; Fleming et al., 2015; Gebraad et al., 2014; Fleming et al., 2016). The implementation of induction control relies on power down-regulation (typically by increasing the pitch angle below rated, i.e., pitch-based AWC) which, as proven technology, is perceived as risk-free in terms of loading. Yaw-based AWC, however, requires operation

with intentional yaw misalignment which has much more pronounced implications on the structural loads turbines are not designed to operate this way (Boorsma, 2012; Kragh and Hansen, 2013; Fleming et al., 2013, 2015; Damiani et al., 2018; Ennis et al., 2018). Even though the risk perception associated to yaw-based AWC is higher, it is at present well recognized by the community that this strategy has much higher potential in terms of energy gain compared to induction control.

In (Kanev et al., 2018), the potential benefits of AWC in terms of lifetime power production and lifetime fatigue loading for

different real-life wind farms were studied through simulations. With respect to power production, it was concluded that the yearly power gains with yaw-based AWC are generally higher than those for pitch-based AWC. One of the conclusions from that study was that, next to power gain, yaw-based AWC can actually result in lower fatigue loads over the lifetime of the wind farm. The loads analysis was performed using a LUT containing loads under various environmental and operational conditions. The LUT approach is based on "gridding", i.e., performing simulations only for a subset of the actual range of environmental

and operational conditions that can occur. This significantly reduces the amount of aeroelastic simulations required to cover all possible inflow and yaw conditions and brings the total number of simulations down to an acceptable number. It should be pointed out that there exist alternative, computationally cheaper, approaches to assessing the farm loads, such as the Frandsen model (Frandsen, 2007), which is recommended in the international standard IEC 64100-1 edition 3. According to the Frandsen model, an effective turbulence level is calculated for the specific site and the wind turbine is simulated for that specific reference

turbulence. The effective turbulence depends on the ambient turbulence and the farm layout and inter-turbine distances. Such an approach, however, is not effective for accurate assessment of the impacts of AWC on the turbine loads, because these impacts are directional and can't be easily translated into impacts on the effective turbulence intensity. For that reason, the LUT approach is followed here.

The LUT contains the fatigue loads and statistics from a large number of aero-elastic simulations with different wind speeds,

turbulence intensities, wake profiles (wake deficit width, depth, and location with respect to the rotor), yaw misalignments, and pitch angle offsets. For given inflow conditions in front of a specific wind turbine in a wind farm, calculated using a wake model as, e.g., FarmFlow (Özdemir and Bot, 2018), the loads on the turbine's components are interpolated from the loads LUT. This LUT-based approach is very attractive as it saves a huge amount of computational time when constructing predictions of the lifetime fatigue loads for each individual turbine for a specific site. Moreover, it enables including the fatigue loading into the

AWC optimization.



To increase the confidence level of the loads LUT approach, it needs to be properly validated. That is the purpose of this work, which has the following objectives:

1. Evaluate if interpolation of the loads in the LUT is an accurate enough method for predicting the loads for conditions that are not present in the LUT.

2. Evaluate the accuracy of the loads calculated using the conventional aeroelastic simulations, using Blade element momentum (BEM) theory. Complex turbine conditions that result from large yaw misalignments violate the assumptions of BEM and its usual correction models.

3. Validate the predictions with respect to wake-induced loads, which are very pronounced load contributors in wind farms.

4. Evaluate if the LUT loads model predictions can be generalized for different turbine scales.

To this end, a series validations studies have been performed based on detailed simulations, wind tunnel measurements and full-scale field tests. These studies are outlined in Sections 3, 4 and 5, respectively. The paper continues in the next section with a detailed explanation of the loads modeling approach, and concludes in Section 6 with some final remarks.

## 2   Farm loads modeling approach

This section describes the wind farm modelling used in this study, as well as the LUT table approach to fatigue loads modeling.

### 2.1   Wind farm model

The wind farm model used in this study is FarmFlow (Özdemir and Bot, 2018; Bot, 2015), which has been developed by ECN/TNO based on the UPMWAKE code (Crespo and Hernández, 1989). It is a 3D parabolised Navier-Stokes code, using a $k - \epsilon$ turbulence model to account for turbulent processes in the wake. The ambient flow is modelled in accordance with the method of Panofsky and Dutton (1984). The free stream wind as a function of height is calculated for a prescribed ambient

turbulence intensity and Monin-Obukhov length, which takes the atmospheric stability into account. The parameters of the $k - \epsilon$ turbulence model are adjusted such that the free stream turbulent kinetic energy matches the value from Panofsky and Dutton for neutral conditions.

The wake model has been improved in van der Pijl and Schepers (2006). Thereto, the parabolization (and the subsequent enormous reduction in computational cost) was retained but the stream wise pressure gradient is not neglected anymore but

prescribed as a source term in the flow equations. The stream wise pressure gradients are calculated via an inviscid, axisymmetric, free vortex wake method. The rotor is assumed to be a uniformly loaded actuator disc. From a prescribed thrust curve of the wind turbine the average axial induction is calculated according to blade element momentum (BEM) theory. The free vortex wake model then calculates the initial induced wake velocities that match the averaged axial velocity deficit in the rotor plane. With this method, the pressure gradients are a function of the axial force coefficient only. To save computational effort,

the pressure gradients are calculated a priori for a large number of axial induction factors, so that the wake model only needs to



interpolate the pressure gradients between the two nearest induction factors in this database. This hybrid method of wake modelling in the near wake region, including an adapted near wake turbulence model, gives very accurate results in an acceptable amount of computational time.

The FarmFlow model supports simulations with active wake control (AWC), allowing that each turbine is operated at a
different power and thrust coefficient (induction control), or with different yaw-misalignment (wake redirection). Implementation of induction control in FarmFlow is rather straightforward by applying different power and thrust curves for induction control. The implementation of the wake redirection control is more complicated and is described below. Since FarmFlow uses prescribed axial and radial pressure gradients in the near wake region in order to induce the wake, i.e. the deceleration and expansion of the flow behind the rotor, implementation of yaw-misalignment is realized by prescribing these pressure gradients
with respect to the yaw angle instead of the flow direction. The effect of this deflection is validated from measurements in a scaled wind farm and with wind tunnel measurements (Bot, 2015). Two empirical correction factors were used to optimize the wake deflection angle and wake deficit values. In addition, the width of the wake is reduced by a factor $\cos(\gamma)$, $\gamma$ being the misalignment angle. A power reduction factor of $\cos\gamma^{1.43}$ is used in agreement with recent measurement studies on full scale wind turbines Fleming et al. (2017).

## 2.2  Loads modeling

In the previous subsection, the wind farm model FarmFlow was summarized. In this section, a loads module is described that enables the estimation of the loading on each turbine at a number of locations. This allows to evaluate the effect of AWC on the turbine loads. Besides analysis, the loads module enables to include the loads into the AWC optimization. The loads module consists of pre-calculated database, constructed using detailed aeroelastic simulations with the software tool Focus/Phatas. The
simulations are performed with a single fictive wind turbine model in the 4MW range operating in a wake situation that cover the entire a wide range of operating conditions which the turbine can encounter during operation in a wind farm. To keep the computational load manageable, only single bell-shaped wake profiles are considered. In practice, a turbine can experience more complex wake situations resulting from wakes from multiple turbines. However, studies with several offshore wind farms with different types of layouts indicated that, in such situations, one of the wakes hitting the rotor strongly dominates the other
one(s) in terms of wake deficit. A situation where a turbine gets two equally strong wakes at both sides of its rotor is, clearly, difficult to imagine as that would imply two upstream turbines to be located at the same distance upstream, and therefore they should stand next to each other.

The operating conditions which have been simulated with Focus/Phatas consist of combinations of the following parameters (see Figure 1 for visualization of these parameters):

– wind speed: due to complex dependency of the loads on the wind speed, a fine grid of points is selected: 4, 6, 8, 10, 12, 14, 16, 18, 20, 22, 25 [m/s]

   – turbulence intensity: the relationship between loads and turbulence intensity is nearly linear for the whole range of wind speeds. Nevertheless, three values (instead of just two) are chosen due to the very pronounced impact of this parameter



on the loads. Based on FarmFlow studies with five different existing offshore wind farms, the following grid points are selected to cover the typical range of variations of this parameter: 5, 15, 30 [%]. Values below 5% are unrealistic in real-life environments.

- – wake deficit width: this parameter can become larger in a multiple wake situation, but a value higher than 3D (D denoting the size of the rotor diameter) is considered unnecessary in the LUT since the simulation space in front of a wind turbine in FarmFlow is around 4D. Wakes with width smaller than 1.2D don't make sense either. This motivates the following choice of grid points: 1.2, 1.8, 3 [D]. The middle value (1.8) is chosen closer to the lowest value (1.2) to ensure higher resolution when the wake effects are more localized.

- – wake deficit depth, relative to free stream velocity. From FarmFlow simulations with several farms it is concluded that this parameter typically remains below 0.58 even for very densely populated farms. In addition, experience with Focus/Phatas calculations occasionally result in numerical problems when too high wake depth in combination with high turbulence intensity is chosen, as locally the wind velocities can become negative. Therefore, the wake deficit depth is topped at 0.5 in the choice for grid points for this parameter: 0, 0.3, 0.5 [-]. The lowest value (zero) corresponds to free-stream operation (no wake), in which case the choice for other wake parameters (wake deficit width, wake location) are immaterial.

- – wake location with respect to rotor: obviously and important parameter, requiring sufficiently fine grid to model its rather nonlinear relationship with the turbine loads. A maximum absolute value larger than 1.5D makes little sense even when the largest considered wake deficit width of 3D is taken. The following grid is selected for the LUT: -1.5, -0.9, -0.6, 0, 0.6, 0.9, 1.5 [D]

- – yaw misalignment angle: misalignments above 30-40% are considered unrealistic in a practical implementation, therefore the following grid is selected: -40, -30, -20, -10, 0, 10, 20, 30, 40 [deg]

- – pitch angle offset: not relevant for this study, but included into the LUT to enable load analysis under induction control deg, motivating the following choice for grid points: 0, 1, 2, 3, 4, 5 [deg]

For each combination of these wake parameters, normal production simulations for six different wind realizations (seeds) have been performed. The simulations are performed with complete three-dimensional wind field that is generated to match the selected values for the wake parameters. This results in a total number of 673596 cases, which were subsequently reduced to 100926 simulations by skipping unnecessary and duplicate cases, such as yaw misalignments and pitch offsets at above rated wind speeds (where AWC will not operate) or different wake widths and locations for zero wake depth (implying no wake at all). The simulations took several days of computation time on a moderate sized computer cluster of about 150 cores. The results from all these simulations are stored into a LUT that comprises the loads database module. The lookup table contains, for each simulated scenario, the calculated fatigue loads and statistics (min, max, mean and std) at a number of locations on the turbine (blades, drive-train, tower, bearings). For the simulations under yaw misalignment it needs to be pointed out that, even



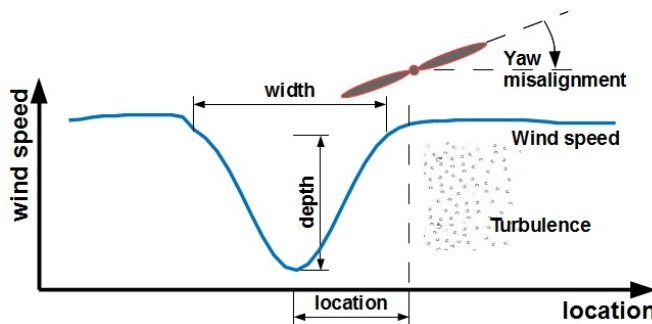

**Figure 1.** Visualization of the parameters used for describing the wake conditions in front of a turbine

though the underlying BEM theory is generally considered as inaccurate under yawed conditions, recent studies (Boorsma et al., 2016) indicate that it captures the load *trends* sufficiently well. Furthermore, since the application here is focused on the analysis of the impact of AWC on the fatigue loads , the primary interest lies in the ability to accurately estimate the relative loads (i.e., the loads increase or decrease) with AWC as compared to the reference/nominal loads without AWC. For

that purpose, using standard BEM theory seems sufficient.

During a farm simulation, FarmFlow determines the wake conditions in front of each turbine, from which the above-listed parameters of a single bell-shaped wake are approximated using least-squares fitting. These wake parameters are subsequently used as input to the loads database to interpolate the corresponding loads on locations. By doing this for the whole range of relevant ambient wind conditions (wind speeds, wind directions, turbulence intensities), and given the corresponding distributions,

the lifetime fatigue loads are calculated for each component at each turbine in the farm.

## 3   Validation by simulations

In this section, validation by simulations is performed. Firstly, in the next section the interpolation properties of the LUT load database are studied using conventional BEM simulations. In the section that follows, higher fidelity simulations are used to assess the prediction capabilities of the LUT approach with respect to yaw-induced loads.

### 3.1   Validation of the interpolation properties of the model

The focus of this section is to evaluate if linear interpolation using the LUT load database is a suitable method for determining the fatigue loads of wind turbines. This would be the case if the LUT database is sufficiently populated which, therefore, is what will be essentially evaluated here. For this purpose, Focus/Phatas aeroelastic simulations were performed for a number of operational conditions, listed in Table 1, that differ from those in contained in the LUT, and the resulting loads are compared

against the predictions from the LUT.

The results from these comparisons are shown in Figure 2, which depicts the fatigue loads from the simulation and LUT interpolation predictions. The loads are compared for the following components: tower bottom resultant, tower top resultant,





**Table 1.** Wake parameters for validation of the interpolation properties of the farm modeling appraoch

| Case | Wind speed [m/s] | Turbulence [%] | Wake depth [-] | Wake width [D] | Wake location [D]) | Yaw error [deg] |
|---|---|---|---|---|---|---|
| 1 | 5 | 0.08 | 0.15 | 1.40 | -0.30 | 25 |
| 2 | 5 | 0.11 | 0.11 | 1.20 | -1.20 | -25 |
| 3 | 5 | 0.15 | 0.38 | 2.70 | -0.85 | 5 |
| 4 | 5 | 0.18 | 0.43 | 1.30 | 0.80 | 15 |
| 5 | 5 | 0.20 | 0.20 | 2.40 | 1.20 | -15 |
| 6 | 7 | 0.25 | 0.25 | 2.00 | -0.75 | 5 |
| 7 | 7 | 0.18 | 0.35 | 1.20 | -0.75 | -5 |
| 8 | 7 | 0.11 | 0.15 | 1.50 | -0.30 | -15 |
| 9 | 7 | 0.08 | 0.35 | 2.60 | 0.75 | -25 |
| 10 | 9 | 0.10 | 0.15 | 1.40 | -1.20 | 25 |
| 11 | 9 | 0.15 | 0.30 | 2.00 | 0.30 | -15 |
| 12 | 9 | 0.20 | 0.40 | 2.20 | -0.30 | 15 |
| 13 | 9 | 0.25 | 0.45 | 1.60 | -1.20 | -5 |

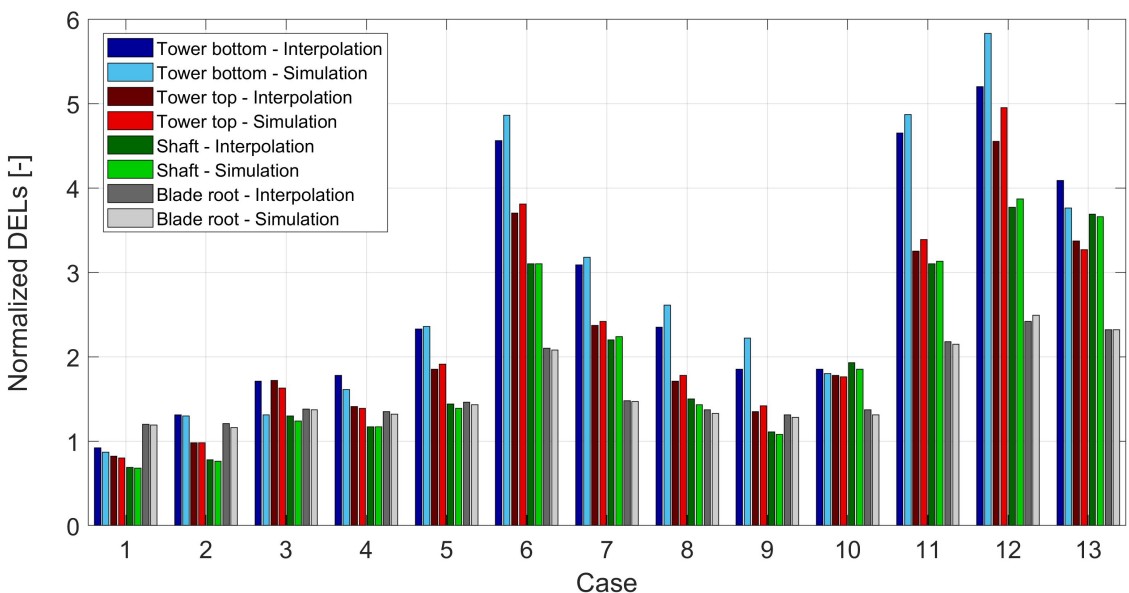

**Figure 2.** Visualization of interpolation and BEM simulation results. When all the parameters are changed at a time, the bigger discrepancies observed for tower at bottom moments





**Table 2.** High-fidelity simulation cases. Different yaw angles are used to evaluate the precision of BEM versus free vortex model

| Case | 1 | 2 | 3 | 4 | 5 | 6 | 7 | 8 | 9 | 10 |
|---|---|---|---|---|---|---|---|---|---|---|
| TI [%] | 5 | 5 | 5 | 5 | 5 | 15 | 15 | 15 | 15 | 15 |
| Yaw [deg] | 0 | 15 | 30 | -30 | -15 | 0 | 15 | 30 | -30 | -15 |

main shaft resultant, and blade root resultant. The loads have been normalized with respect to the loads at 8 m/s wind speed, 5% turbulence, no wake (zero wake depth), zero yaw misalignment and zero pitch angle.

In can be seen from Figure 2 that the loads interpolated from the LUT seem to be generally in good agreement with those from the simulations. Excellent agreement is observed for tower at top, shaft and blade results, while some relatively small

discrepancies are present in the tower bottom loads for some cases. These are primarily attributed to the different wind field realizations used in the construction of the LUT and the simulations performed for this comparison.

## 3.2  Validation by higher fidelity simulations

Next, the precision of the LUT load modeling for yawed flow conditions is studied, as those inherent for wake redirection AWC. As explained in Section 2, the LUT has been constructed using conventional Focus/Phatas simulations in which the

aerodynamics are computed using BEM theory. However, complex turbine conditions that result in non-uniform induction like e.g. yawed inflow, pitch asymmetry, or heavily deflected rotor blades violate the assumptions of BEM and its usual correction models (Boorsma et al., 2016).

For this reason, the Aerodynamic Wind Turbine Simulation Module (AWSM) (Boorsma et al., 2016) has been developed. AWSM code relies on a more sophisticated approach that accounts for the complex flow phenomena on wind turbine rotors:

lifting-line theory in combination with a free vortex wake method. This approach is based on a more physics-based representation, especially for wake-related phenomena, and is more accurate than BEM in predicting the loads induced by oblique inflow. The higher accuracy comes, of course, at the price of much higher computational complexity. This makes the application of free vortex wake models, such as AWSM, for the construction of the loads LUT table prohibitive at present.

In this study, the yaw-induced fatigue loads from AWSM simulations are compared to those from BEM simulations. To this

end, AWSM and BEM simulations are performed using DTU 10 MW reference wind turbine Bak et al. (2013). Notice the much larger scale of this wind turbine than that of the turbine used for building the LUT database. Hence, besides the load prediction accuracy under oblique inflow, the scalability of the LUT model will be indirectly tested as well.

In the simulations, turbulent inflow at 8 m/s was used in combination with different turbulence intensities and yaw misalignment angles, as listed in Table 2. Turbulence intensities of 5% and 15% are considered, as could be encountered in a free stream

and waked operation of a wind turbine in a wind farm.

In Figure 3, the blade root resultant damage equivalent loads from the AWSM and BEM simulations are compared. The loads have been normalized with respect to the BEM loads at zero yaw misalignment and 5% turbulence intensity. From the depicted results, a few observations can be made.

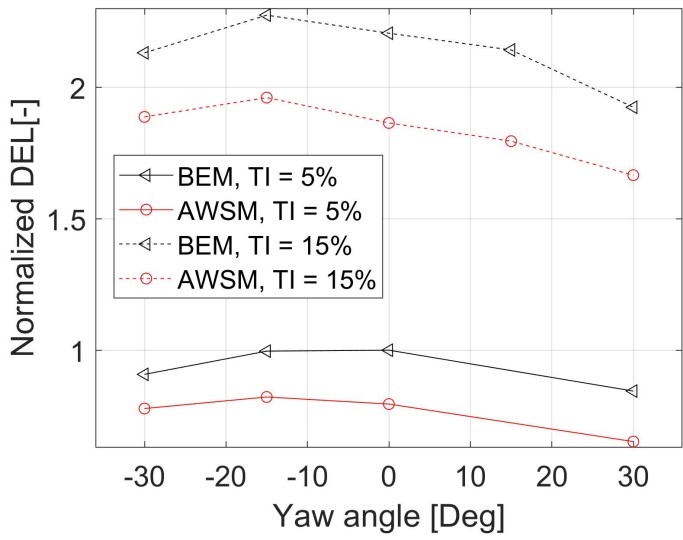

**Figure 3.** Normalized blade root DEL as a function of yaw angle for the AWSM and BEM models

– The higher turbulence intensity results in higher fatigue loads, both for BEM and AWSM. Moreover, by moving from 5% to 15% turbulence, the relative load increase is comparable for both models.

– The relative load changes due to misalignment are much smaller than those due to turbulence. The results here suggest that yaw misalignment can increase the turbine loads in the range of 10-15%, but the impact of wakes on the loads are much more pronounced. Due to the higher turbulence intensity in the wake, a turbine downstream may experience up to 250% higher loading then that in free stream conditions. This implies that wake-induced loading is more pronounced that yaw-induced loading. Thus, yawing the upstream turbine to decrease the turbulence in the downstream machines will lead to loads alleviation of the farm as a whole. This fact is seen as the reason that wake redirection control can result in lower fatigue loading, see Kanev et al. (2018).

– Taking the zero yaw angle as reference, the load trends are generally well captured by BEM for both positive and negative misalignments. This implies that in terms of relative loads impacts by AWC, the BEM-based LUT approach seems suitable.

– In absolute sense, BEM significantly overpredicts the loads as compared to AWSM. This is completely in line with earlier findings by Boorsma et al. (2016); Boorsma (2012). Therefore, using the LUT to predict the lifetime fatigue loads on the turbines in a wind farm can be considered as a conservative, and hence, safe approach.



**Table 3.** Selected load cases from Mexico experiments. $V_{inf}$ is the free stream wind speed, $N_{rotor}$ is the rotational speed of the rotor, $\lambda$ is the tip speed ratio.

| $V_{inf}$ [m/s] | Yaw [°] | Pitch [°] | $N_{rotor}$ [rpm] | $\lambda$ [-] |
|---|---|---|---|---|
| 9.97 | -30 | -2.3 | 425.1 | 10.042 |
| 9.97 | 0 | -2.3 | 425.1 | 10.045 |
| 9.99 | 8 | -2.3 | 425.1 | 10.030 |
| 10.04 | 15 | -2.3 | 425.1 | 9.977 |
| 9.98 | 30 | -2.3 | 425.1 | 10.038 |

## 4 Validation by wind tunnel experiments

The focus of this section is to validate the LUT load model against wind tunnel measurements under misaligned inflow conditions. This is done using measurements gathered in the New Mexico project (Boorsma and Schepers, 2014) and CL-Windcon project (CL-Windcon).

### 4.1 Mexico wind tunnel experiment

The objective of the New Mexico project (Boorsma and Schepers, 2014) was to create a database of detailed aerodynamic and load measurements on a 4.5m wind turbine model, in a large and high-quality wind tunnel. The selected measurements are the blade forces, and the inflow conditions include different yaw misalignment angles, as shown in Table 3. All considered test cases involve operating the turbine with its nominal pitch angle (-2.3°), rotor speed (425.1 rpm) and tip speed ratio (10). Using the measured forces along the blade span, the blade root out-of-plane fatigue loads are calculated. These are subsequently compared against the predictions form the loads LUT, which are interpolated for the the following inputs: 10 m/s wind speed, 0% turbulence intensity, zero wake depth (in the New Mexico experiment the turbine operates in free stream), yaw misalignments in accordance with the selected cases from the New Mexico experiment (-30°, 0°, 8°, 15°, 30°), and nominal pitch angle. The wake width and location inputs to the LUT (both set equal to zero), are irrelevant due to te zero wake depth. Notice that zero turbulence is outside the range of turbulence intensity values stored in the LUT (see Section 2.2). Since, as already mentioned above, the impact of turbulence intensity on the loads is very pronounced, it was decided to linearly extrapolate the loads for zero turbulence rather than choosing the lowest turbulence values available in the LUT.

Figure 4 depicts the measured blade root out-of-plane loads and the LUT predictions as function of the yaw misalignment, both normalized with respect to the loads at zero yaw. Results show that the load trends (the slopes of the curves) are comparable. For the wind tunnel experiment, the loads are practically symmetric around the zero-degree yaw. On the other hand, the LUT loads prediction achieves its minimum value at a positive, non-zero yaw angle. This is consistent with the results in many recent studies, see e.g. Ennis et al. (2018); Kragh and Hansen (2013); Boorsma (2012). The asymmetry of the LUT loads curve with respect to the zero-degree yaw is due to the presence of vertical wind shear in the calculation of the LUT loads.





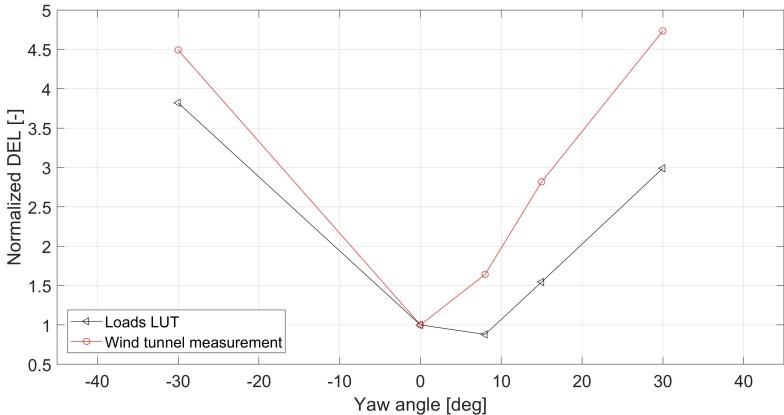

**Figure 4.** Normalized blade root out-of-plane loads: Mexico wind tunnel measurements vs. LUT batabase prediction

**Table 4.** CL-Windcon single-turbine experiment cases

| Case | Wind speed [m/s] | TI [%] | yaw [°] |
|------|------------------|--------|---------|
| 1 | 5.7 | 5 | -40:10:40 |
| 2 | 5.7 | 10 | -40:10:40 |

The shear counteracts the advancing and retreating blade effect for positive yaw, leading to lowest loads at some positive yaw angle. In the New Mexico experiment there is no wind shear, which explains the fact that the lowest load appears at zero yaw misalignment. Therefore, it can be concluded that the LUT loads modeling seems a viable approach for predict the relative impact of wake redirection on the loads of intentionally misaligned wind turbines.

## 4.2 CL-Windcon wind tunnel experiments

In the CL-Windcon experiments (CL-Windcon, 2017), the first wind tunnel entry involved a series of tests on a single wind turbine model operated with different yaw misalignment angles and power setpoints. The recorded time series of the tower base fore-aft moment are used to calculate the corresponding DEL, which are subsequently compared to the LUT load predictions. In the CL-Windcon wind tunnel experiments, spires are used to generate vortexes at the beginning of the test section, and bricks were placed on the ground to represent surface roughness. As a result, two boundary layers were created, one for low turbulence intensity (5%) and one for high turbulence intensity (10%). The wind tunnel conditions, reported in Table 4, are used as inputs to the LUT to interpolate the tower base fore-aft moment and compare it to the measurement. Given that the turbine model is in free stream condition, the wake depth input is set to zero (no wake).

The results are shown in Figure 5, comparing the measured tower bottom fore-aft fatigue load to the LUT prediction for turbulence of 5% (left plot) and 10% (right plot). The loads are normalized against the load at zero-yaw and 5% turbulence.





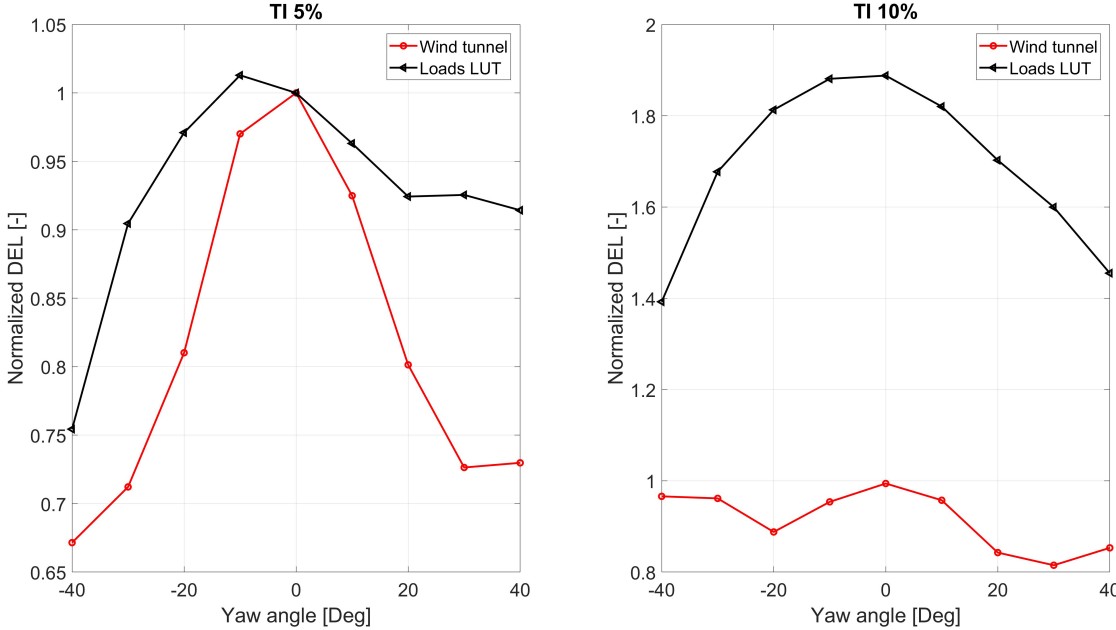

**Figure 5.** CL-Windcon and loads module tower base fore-aft moment as a function of yaw, for low turbulence (left) and high turbulence (right)

In the low turbulence case, a similar trend is observed between the LUT prediction and the tunnel experiment. The database overestimates the loads and, more importantly, the tower loads decrease with yaw misalignment. For high turbulence, however, a big discrepancy is observed with respect to the effect of the turbulence intensity on the loads: due to the much higher turbulence, the LUT load predictions are much higher, while the wind tunnel measurements do not share this trend. Further
5 analysis of the results indicated that this is due to the inertial loads being the main contributor to fatigue loads. Due to the small scale of the turbine, the tower frequency (around 14 Hz) is well outside the bandwidth of the turbulence excitation. At the same time it is very close to the rotational frequency of the rotor, getting excited by the rotor (aerodynamic and mass) imbalance. This deterministic excitation outweighs by much the impact of the turbulence on the loads. As a result, the turbulence intensity has practically no impact on the tower fatigue loading in this wind tunnel experiment.
10 Due to this, it is concluded that in terms of tower bottom loads these measurements are unrealistic for a real-life modern wind turbine and are therefore considered not suitable for validation of the LUT load model.

## 5 Validation by full-scale measurements

In this section, the LUT load model is compared against full scale field measurements. The measurements are performed at the ECN's Wind Turbine Test Site Wieringermeer (EWTW), the Netherlands. The farm consists of five research turbines which





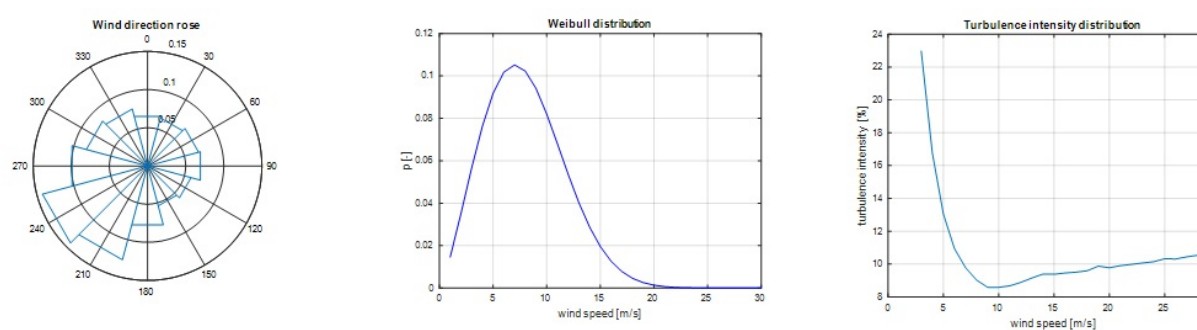

**Figure 6.** Wind conditions at EWTW: (left) wind rose, (middle) Weibull distribution, (right) turbulence intensity as a function of wind speed

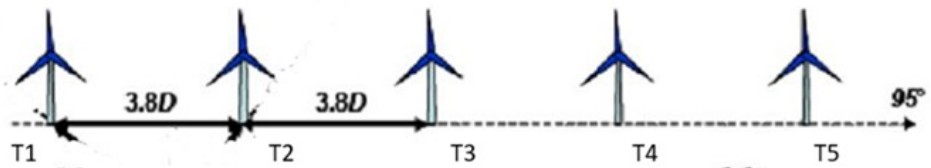

**Figure 7.** EWTW's layout

are oriented in a single line with a mutual distance of 3.8 D (see Figure 7). The orientation of the row is 95°-275°(0 being North). The turbines are variable speed, pitch controlled, and have a rotor diameter of 80 m and hub height of 80 m. The wind conditions at the site are given in Figure 6: the wind direction distribution (left plot), wind speed distribution (middle plot) and turbulence intensity as function of wind speed (right plot). The measurements are obtained during normal operation,

without misalignment, and therefore the focus here is on validation of the wake induced load prediction capability of the LUT approach.

Given the wind turbines and site conditions, a FarmFlow model is built and used to estimate the inflow conditions for each turbine for the whole range of wind speeds and wind direction. These are subsequently used to interpolate the loads from the LUT, as explained in Section 2.2.

For the analysis, ambient wind speeds of 6, 7, 8, 9, 10 m/s are used. The loads on the tower bottom and blade root are measured on the second turbine from the West, turbine T2. All the measured fatigue samples are filtered to match the simulated turbulence conditions. These measurements are compared to the LUT load predictions for the different wind speeds. Furthermore, a normalization is applied based on the loads at wind speed of 8 m/s and wind direction of 180° (free stream).

Since the farm layout consists of a single row of turbines, there are only sectors of wind directions in which the measured

turbine is in wake: around 95° and 275°. In Figures 8-9, the measured loads are compared to the LUT predictions for different wind speeds. The blade root out-of-plane loads (left plots in the figures) and tower bottom fore-aft loads (right plots) are given. The grey dots in the plots represent the raw (filtered and normalized) measurements, while the red solid lines give the binned averages of these data points. The solid black lines depict the normalized LUT load predictions.



**Figure 8.** Comparison of LUT loads prediction to EWTW measurements for 6, 7 and 8 m/s







**Figure 9.** Comparison of LUT loads prediction to EWTW measurements at 9 and 10 m/s

The wind direction sectors for which the measured turbine T2 is in a wake condition are clearly identifiable in Figures 8-9 by the large peaks in the loading on the blades. The magnitude of these two peaks shows generally good agreement between the LUT model and the real-life measurements. This is especially valid for the blade root out-of-plane moments, for which the LUT predictions compare very well with the measurements. With respect to the tower loads, the LUT loads overpredict the measurements in the two sectors of waked operation. It is generally observed that the measured tower loads seem less sensitive to waked inflow conditions than the blade loads. In the near future, full-scale measurements will be performed on turbines with yaw misalignment and operating in a wake situation.



## 6 Conclusions

This paper presented the results of a number of studies focused on the validation of the LUT approach to modeling the loads on turbines in wind farms. The approach represents a computationally attractive way to study the impact of wake redirection AWC on the turbine loads. The validation studies included conventional (BEM) and detailed (free vortex wake) simulations, data from two wind tunnel measurements performed under yaw misalignment, and full-scale field measurements.

The BEM simulations were used to evaluate the interpolation properties of the LUT. The results indicated that, for the chosen resolution of the LUT, the interpolated loads accurately approximate the simulated loads.

The free vortex wake simulations with the AWSM code confirmed earlier findings that the fatigue loads predicted by BEM models tend to significantly overpredict the loads from AWSM simulations. This implies that using BEM models (as those used to construct the LUT) is a conservative, though safe approach to assess the loads on turbines. Another observation, applicable to both BEM and AWSM, the loads are shown to increase significantly for higher turbulence levels. This is also consistent with other results showing the wake-induced loading is much more pronounced than the loading due to misalignment. This is also the main reason that, as discussed in Kanev et al. (2018), wake redirection AWC can reduce the overall lifetime fatigue loading even though for some specific wind conditions the loads on some turbines may increase a bit due to misalignment. Finally, it is seen in the comparison with AWSM that the load trends are generally well captured by BEM for both positive and negative misalignments. This implies that in terms of relative loads impacts by AWC, the BEM-based LUT approach seems suitable.

The wind tunnel experiments proved very useful for validating the yaw LUT prediction of the yaw-induced load. The New Mexico experiment indicated that the sensitivity of the blade out-of-plane loads to changes in the yaw misalignment angle are very well modeled by the LUT approach even though the tunnel test is performed with a much smaller turbine. Interesting observation was that due to lack of wind shear in the tunnel experiment, the lowest blade loading was achieved at zero yaw misalignment, while the present of shear in the simulations used for creation of the LUT resulted in lowest loads at non-zero, positive yaw angle. This is also consistent with previous studies. The CL-Windcon tunnel tests involved experiments with two levels of artificially generated turbulence. Unfortunately, the measured tower loads proved to be very insensitive to variations in the turbulence. The reason for that was that for this scaled turbine model, the main contributor to the tower loads is the relatively high tower frequency, excited primarily by 1p effects due to aerodynamic and/or mass imbalance. These outweighed by much the fatigue loads induced by (low frequency) turbulence. As a result of that, the CL-Windcon measurements were not useful for assessing the accuracy of the wake-induced loads predictions by the LUT, but they did confirm the findings with respect to yaw-induced loading.

Finally, the field measurements on EWTW were compared to the LUT load predictions for a range of wind speeds. The agreement was very good, especially for the blade root bending moments. With respect to tower loads the LUT estimates generally overpredicted the measurements for the wind directions with waked inflow. The measured tower loads were also found less sensitive to variations in the inflow conditions than the blade loads.





*Data availability.* Data is not available due to confidentiality issues.

*Author contributions.* HMR performed all the analysis and prepared a first draft as part of his MSc final project, which he performed at ECN part of TNO. SK supervised HMR on a daily basis, performed simulations, and helped with the analysis and interpretation of the results. SK had a major role in the preparation of the final version of the manuscript. JWvW and BD had an advisory role as formal supervisors from the

5   TU-Delft.

*Competing interests.* The authors declare that they have no conflict of interest.

*Acknowledgements.* Koen Boorsma is acknowledged for the support he provided in setting up the AWSM simulations.





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
