# Peer review of "Validation of a lookup-table approach to modeling turbine fatigue loads in wind farms under active wake control"

_Wind Energy Science, 2019_

## Referee Comment (RC1) · Anonymous Referee #1 · 12 Jul 2019

Page 4 Line 11: What are these corrections? Are they explained in (Bot 2015)?

Page 4 Line 14: Wouldn't it be useful to use the results with yaw misalignment in this paper to validate this power reduction factor, as well as validating the loads?

Page 4 Line 27: Upstream turbines would be next to each other ... perhaps also unlikely, but they could also be at different distances and with different thrust coefficients.

Page 5 Line 4: Please specify how the wake width is defined with respect to the Gaussian profile.

Page 6 Line 6: How far upstream of the rotor do you measure the wind speeds in Farm-

Flow, and are they affected by rotor induction? Is the wind input to Phatas considered to be far upstream, i.e. unaffected by the rotor induction? Are the wind speeds from FarmFlow compatible with what's required for Phatas, and if not, how do you correct for this?

Table 1 caption - spelling of 'approach'.

General comment: Is the LUT assumed to apply to all turbines, or was it recalculated for each turbine used in the various comparisons? If it's considered general, how do you scale the loads for different turbine sizes / rotor speeds / rated powers (power densities) etc.? How would you account for turbines with or without individual pitch control, tower and drive-train damping algorithms, and the position of the tower frequency (and other frequencies) with respect to rotational frequency and its multiples? These effects can significantly affect loads. So can the detail of the control design.

Page 10 Line 11: spelling of 'from'.

Section 4.1 - Mexico experiment, Page 11, Line 3: "LUT loads modeling seems a viable approach for predict" (typo "predicting") Why not strengthen this conclusion by running comparative BEM calculations without wind shear for these conditions?

Page 11, Line 9: plural of 'vortex' is 'vortices'

Section 5 Validation by full-scale measurements: For the tower, agreement on magnitude is actually poor, especially in low winds, and especially in the waked sectors. This is perhaps not so surprising at 90 deg since T2 is then in a multiple-wake situation - this might also explain the discrepancies in the blade loads. Can any other explanations be suggested for the tower load discrepancies?

---

## Author Comment (AC1) · 9 Aug 2019

Dear reviewer, thank you for the evaluation of our manuscript and your useful comments. We have carefully considered them all in the revised version of the paper, as explained below. The changes made to the manuscript have been marked in yellow in the revised version for your convenience.

Your comment: Page 4 Line 11: What are these corrections? Are they explained in (Bot 2015)?

Response: We have clarified these corrections in the revised manuscript, see page 4,

lines 12-17.

Your comment: Page 4 Line 14: Wouldn't it be useful to use the results with yaw misalignment in this paper to validate this power reduction factor, as well as validating the loads?

Response: The full-scale tests used in the study are done without yaw misalignment, so these data cannot be used to validation. The wind tunnel tests on the other side are performed with much smaller turbines in rather idealistic inflow conditions, and as such are considered not realistic for verifying the power loss factor. In our experience with a few other commercial wind turbines of similar size this factor is close to the one reported by NREL in the paper in question (we added this in the text, see marked lines 19-20 on page 4). Of course, for obtaining optimal results with AWC optimization, this factor should ideally be determined using measurements on a case-to-case basis. However, validation of this parameter goes outside of the scope of this paper, which is focused on the load modeling.

Your comment: Page 4 Line 27: Upstream turbines would be next to each other ... perhaps also unlikely, but they could also be at different distances and with different thrust coefficients.

Response: Yes, good point indeed. Of course, cases can be constructed which give rise to wakes in front of a turbine which are not well approximated with a bell-shaped profile. However, most of the cases we could imagine we considered not realistic for the majority of the real-life offshore wind farms. Extending the loads database to model, for instance, double-bell shaped wake profiles would have given rise to a significantly larger amount of aeroelastic simulations necessary to populate the database. It was therefore decided that the resulting increase in computational complexity does not weigh against the expected added value in practice, and therefore it was decided to keep the simulations limited to single bell shaped wakes. We added some additional words to clarify this point better in the text, see the marked text on page 4 last line to

page 5 line 4.

Your comment: Page 5 Line 4: Please specify how the wake width is defined with respect to the Gaussian profile.

Response: A mathematical formulation of the wake profile has been included, see marked text on page 5, lines 5-10.

Your comment: Page 6 Line 6: How far upstream of the rotor do you measure the wind speeds in Farm- Flow, and are they affected by rotor induction? Is the wind input to Phatas considered to be far upstream, i.e. unaffected by the rotor induction? Are the wind speeds from FarmFlow compatible with what's required for Phatas, and if not, how do you correct for this?

Response: The inflow in front of each wind turbine, used in FarmFlow is undisturbed by the rotor induction. Similarly, the wind input to Phatas is generated in the same way (undisturbed by rotor). There are therefore no significant differences between the properties of the incoming airflow in FarmFlow and Phatas. We have now clarified this point in the text on page 7, lines 3-5.

Your comment: Table 1 caption - spelling of 'approach'.

Response: Typo corrected (page 7, Table 1 caption)

Your comment: General comment: Is the LUT assumed to apply to all turbines, or was it recalculated for each turbine used in the various comparisons? If it's considered general, how do you scale the loads for different turbine sizes / rotor speeds / rated powers (power densities) etc.? How would you account for turbines with or without individual pitch control, tower and drive-train damping algorithms, and the position of the tower frequency (and other frequencies) with respect to rotational frequency and its multiples? These effects can significantly affect loads. So can the detail of the control design.

Response: Yes, valid comment of course. The LUT database is created with a wind

turbine model and controller according to the current "common practice". As such, it may not be representative for specific cases such as wind turbines with soft-soft towers, low induction rotors, and advanced control algorithms including IPC, tower damping, LiDAR-based control etc. For the more standard cases, the results from this paper suggest that the LUT approach is suitable for different wind turbine types when it comes to predicting the load trends (making it possible to judge whether under AWC loads increase or decrease, and by how much), rather than the absolute loads. This comment is now added at the end of the Conclusions section (see last paragraph).

Your comment: Page 10 Line 11: spelling of 'from'.

Response: Typo corrected (page 10, line 14)

Your comment: Section 4.1 - Mexico experiment, Page 11, Line 3: "LUT loads modeling seems a viable approach for predict" (typo "predicting") Why not strengthen this conclusion by running comparative BEM calculations without wind shear for these conditions?

Response: Good point. Comparison of the Mexico measurements to BEM calculations without shear has already been done in the past in Boorsma (2012). The results there indicate that the conventional BEM implementation in Focus/Phatas give a rather good prediction of the load trends due to misalignment, and confirm the findings here. Also, the lowest loads at non-zero misalignment is confirmed by the simulations. We added a reference to that in the text (page 11, line 14).

Your comment: Page 11, Line 9: plural of 'vortex' is 'vortices'

Response: Corrected.

Your comment: Section 5 Validation by full-scale measurements: For the tower, agreement on magnitude is actually poor, especially in low winds, and especially in the waked sectors. This is perhaps not so surprising at 90 deg since T2 is then in a multiple-wake situation – this might also explain the discrepancies in the blade loads. Can any other

explanations be suggested for the tower load discrepancies?

Response: Yes, good point. We added the following text at the end of Section 5 (bottom of page 14):

"With respect to the tower loads, the LUT loads predictions are not good, especially for the lower wind speeds. It is observed that the measured tower loads seem quite insensitive to waked inflow conditions. This observation is similar to the one made in Section 4.2 for the wind tunnel experiments, where the inertial loading due to rotor imbalance was suggested as the possible cause for this. Since he wake effects on the loading are clearly seen in the blade loads here, significant rotor imbalance seems like a plausible reason here as well. However, time series data was not readily available to verify this. In the near future, new full-scale measurements with another turbine type will be performed on turbines with and without yaw misalignment and operating in a wake situation, which is expected to give new insights and further validate/improve the LUT load modelling approach."

Once again, thank you for your fruitful comments which we hope to have considered appropriately in the revised version.

Best regards, Stoyan Kanev

Please also note the supplement to this comment:
https://www.wind-energ-sci-discuss.net/wes-2019-34/wes-2019-34-AC1-supplement.pdf

---

## Referee Comment (RC2) · Anonymous Referee #2 · 14 Aug 2019

The paper makes several important contributions to the wind farm control and wake redirection literature with respect to understanding load impacts. The paper includes very useful analysis and validation, and the proposed LUT-based method for understanding load impacts is compellingly presented and analyzed. Introduction is well written and connection to the literature is good, the paper well explains the contributions of the paper. Believe will be a very useful paper for the field.

Some technical comments follow:

Main general comment:

Could you explain a little on the selection of loads analyzed, is there consensus opinion

that this particular set of loads well covers/correlates all loads? For example, one might expect yaw bearing loads to be particularly impacted by offsetting wake, but is this essentially included in the tower top load? 1-2 paragraphs on why the included loads were selected, and if all excluded loads can be expected to behave similarly would be much appreciated.

Small technical comments

Page 5: You remove offsets at above rated wind speeds where AWC will not operate, but I believe AWC could well operate effectively up to wind speeds where downstream turbines are rated, which would be above-rated for upstream by 1-2 m/s, was this accounted for?

Fig 2: Would be helpful to also include what is the normalization in the caption

Page 9: "but the impact of wakes on the loads are much more pronounced", I believe you, but can this be inferred from Fig 3?

Fig 4: Is this somehow inverted on the y-axis with respect to fig 3? I understood the discussion in the text on the location of the nadir, but I was confused on the inversion

Fig 8: Is raw data 10-minute bins? The agreement is nice

---

## Author Comment (AC2) · 15 Aug 2019

Dear reviewer, thank you for the evaluation of our manuscript and your useful comments. We have carefully considered them all in the revised version of the paper, as explained below. The changes made to the manuscript have been marked in yellow in the revised version for your convenience. Please notice that this revised version contains also the changes made to the manuscript based on the comments from the other reviewer. We have decided to leave those unmarked to avoid confusion with the changes made based on your comments.

Your comment: Could you explain a little on the selection of loads analyzed, is there

consensus opinion that this particular set of loads well covers/correlates all loads? For example, one might expect yaw bearing loads to be particularly impacted by offsetting wake, but is this essentially included in the tower top load? 1-2 paragraphs on why the included loads were selected, and if all excluded loads can be expected to behave similarly would be much appreciated.

Response: The locations at which the loads have been stored in the LUT have been carefully selected in consultation with experts from the industry. The goal was to choose a limited number of critical locations that are representative for the complete turbine structure. In the revised manuscript we have added Table 1 (page 7) which gives an overview of the complete set of locations. In addition, we have added a few lines to clarify this selection, see lines 24-29 on page 7.

Your comment: Page 5: You remove offsets at above rated wind speeds where AWC will not operate, but I believe AWC could well operate effectively up to wind speeds where downstream turbines are rated, which would be above-rated for upstream by 1-2 m/s, was this accounted for?

Response: This statement was misleading, the offsets are included in the LUT up to 14 m/s, where 12 m/s is the rated wind speed for the turbine. We have clarified this point in the revised version on page 6, lines 21-22.

Your comment: Fig 2: Would be helpful to also include what is the normalization in the caption

Response: The applied normalization is included in the caption now (page 9)

Your comment: Page 9: "but the impact of wakes on the loads are much more pronounced", I believe you, but can this be inferred from Fig 3?

Response: We agree that this was not explained clearly enough. The figure shows that the loads increase due to yawing is not more than around 10-15%, but the loads increase due to higher turbulence is much larger (>250%). Since the turbulence in

the wake is higher than in free stream, with 15% being more or less representative for a single wake situation offshore, and 5% for free stream, the loads experience at a downstream turbine due to wake effects (and hence higher turbulence) will be in the order of 250% - much more pronounced that the loads increase of 10-15% due to yawing. It is based on this fact that we argue that wake-induced loading is more pronounced that yaw-induced loading. We have revised the respective part of the text in the manuscript to better explain our argumentation, please refer to page 10 line 22 to page 11 line 1 in the revised manuscript.

Your comment: Fig 4: Is this somehow inverted on the y-axis with respect to fig 3? I understood the discussion in the text on the location of the nadir, but I was confused on the inversion

Response: No, no inversion. The reason for the different shape of the curves here is simply the lack of any turbulence. In the case of lack of turbulence, the effect of yaw offset on the loads is essentially exaggerated. However, adding just a little turbulence already induces loads that are more pronounced than those due to yaw misalignment only, so that at 5% turbulence the curves take the shape of those in Fig 3 and Fig 5. We have added some text to clarify this in the revised manuscript, see page 13 lines 6-9.

Your comment: Fig 8: Is raw data 10-minute bins? The agreement is nice

Response: Yes indeed, time series of duration 10 minutes have been used to calculate the raw loads. We added a clarification (page 15 line 9).

Once again, thank you for your fruitful comments which we hope to have considered appropriately in the revised version.

Best regards,

Stoyan Kanev (on behalf of all authors)

[Figure]

Please also note the supplement to this comment:
https://www.wind-energ-sci-discuss.net/wes-2019-34/wes-2019-34-AC2-supplement.pdf

[Figure]

**Supplement:**

[revised manuscript text omitted]